# Protein Transduction Domain-Mediated Delivery of Recombinant Proteins and In Vitro Transcribed mRNAs for Protein Replacement Therapy of Human Severe Genetic Mitochondrial Disorders: The Case of Sco2 Deficiency

**DOI:** 10.3390/pharmaceutics15010286

**Published:** 2023-01-14

**Authors:** Androulla N. Miliotou, Parthena F. Foltopoulou, Alexandra Ingendoh-Tsakmakidis, Asterios S. Tsiftsoglou, Ioannis S. Vizirianakis, Ioannis S. Pappas, Lefkothea C. Papadopoulou

**Affiliations:** 1Laboratory of Pharmacology, School of Pharmacy, Faculty of Health Sciences, Aristotle University of Thessaloniki, 541 24 Thessaloniki, Macedonia, Greece; 2Department of Health Sciences, KES College, Nicosia 1055, Cyprus; 3Department of Life and Health Sciences, University of Nicosia, Nicosia 1700, Cyprus; 4Laboratory of Pharmacology and Toxicology, Faculty of Veterinary Science, University of Thessaly, 431 00 Karditsa, Thessaly, Greece

**Keywords:** protein replacement therapy, mitochondrial disorders, protein transduction domain technology, recombinant proteins, in vitro transcribed mRNA (IVT-mRNA), Sco2

## Abstract

Mitochondrial disorders represent a heterogeneous group of genetic disorders with variations in severity and clinical outcomes, mostly characterized by respiratory chain dysfunction and abnormal mitochondrial function. More specifically, mutations in the human *SCO2* gene, encoding the mitochondrial inner membrane Sco2 cytochrome *c* oxidase (COX) assembly protein, have been implicated in the mitochondrial disorder fatal infantile cardioencephalomyopathy with COX deficiency. Since an effective treatment is still missing, a protein replacement therapy (PRT) was explored using protein transduction domain (PTD) technology. Therefore, the human recombinant full-length mitochondrial protein Sco2, fused to TAT peptide (a common PTD), was produced (fusion Sco2 protein) and successfully transduced into fibroblasts derived from a *SCO2*/COX-deficient patient. This PRT contributed to effective COX assembly and partial recovery of COX activity. In mice, radiolabeled fusion Sco2 protein was biodistributed in the peripheral tissues of mice and successfully delivered into their mitochondria. Complementary to that, an mRNA-based therapeutic approach has been more recently considered as an innovative treatment option. In particular, a patented, novel PTD-mediated IVT-mRNA delivery platform was developed and applied in recent research efforts. PTD-IVT-mRNA of full-length *SCO2* was successfully transduced into the fibroblasts derived from a *SCO2*/COX-deficient patient, translated in host ribosomes into a nascent chain of human Sco2, imported into mitochondria, and processed to the mature protein. Consequently, the recovery of reduced COX activity was achieved, thus suggesting the potential of this mRNA-based technology for clinical translation as a PRT for metabolic/genetic disorders. In this review, such research efforts will be comprehensibly presented and discussed to elaborate their potential in clinical application and therapeutic usefulness.

## 1. Introduction to Mitochondrial Disorders

### 1.1. Mitochondria: Characteristics and Function

It has been over 150 years since the discovery of the mitochondrion, the double-membrane intracellular organelle with critical roles in multiple cellular and metabolic pathways [1,2]. Still, the interest for medicinal applications targeting deficiency of their function has mostly been undertaken for the last three (3) decades.

These essential semi-autonomous organelles consist of two membranes, i.e., the outer (OMM) and the inner (IMM) mitochondrial membrane, separated by a small intermembrane soluble space (IMS). The two membranes have different architecture and permeability properties, where the OMM allows the passage of small proteins and ions, while the IMM is more restricted towards permeabilization. The IMM folds over many times and creates layered structures called cristae, which extend into the interior soluble compartment (the matrix) of the organelle [3,4]. The main function of mitochondria in eukaryotic cells is the production of chemical energy through the oxidative phosphorylation (OXPHOS) pathway [5,6]. This metabolic energy-generating pathway is embedded in the inner mitochondrial membrane and under normal physiological conditions produces more than 90% of the cellular energy. Moreover, in the matrix, metabolic pathways such as the β-oxidation of fatty acids, the tricarboxylic acid cycle (TCA or citric acid cycle or Krebs cycle), as well the urea cycle take place. Other cellular processes in which mitochondria are implicated include the initiation of apoptosis (programmed cell death) [7,8], calcium homeostasis [9], heme and iron–sulfur cluster biosynthesis [10], amino acid and lipid metabolism [11], and generation of reactive oxygen species (ROS) [12,13].

Mitochondria are highly dynamic organelles with their own set of DNA, called mitochondrial DNA (mtDNA), of which each mitochondrion in humans can have two to ten copies of circular DNA. It is widely believed that these organelles were once primitive bacteria with their own DNA that were swallowed up by larger cells more than a billion years ago. As these bacteria and their host cells evolved, they developed a co-dependent relationship [14]. Through evolution, many of the mitochondrial genes either were lost or transferred to nucleus [15]; thus, the mitochondrial proteome is derived from both mitochondrial and nuclear DNA (nDNA). mtDNA is a highly compact, circular, double-stranded, haploid DNA strand molecule of 16,569 base pairs (bp) in humans, lacking introns and containing 37 genes that encode 13 structural subunits of the OXPHOS system (complexes I, III, IV, and V) and 22 tRNAs necessary for intramitochondrial protein synthesis and the small (12S) and large (16S) ribosomal RNAs (rRNAs) [16]. The organization of the mammalian mitochondrial genome is highly conserved [17], while only 3% of the mtDNA (the D-loop area of the 1.1 kilobases (kb) in humans) is a non-coding region and contains elements crucial for its replication and transcription [18]. The mitochondrial DNA is replicated, transcribed, and translated within the matrix space, with a modified mitochondrial genetic code differing from the universal one in four codons [19], and it is maternally inherited [20].

The number of mitochondria in cells varies between organisms, tissues, and cell types and depends mainly on their metabolic needs. These numbers range per cell from a single large mitochondrion to thousands of organelles. In humans, erythrocytes (mature red blood cells) are the only cells that lack mitochondria, while organs with high demands for energy production, such as heart, muscle, liver, kidney, and to a certain extent the brain, can have hundreds or thousands of these organelles in each cell.

### 1.2. Oxidative Phosphorylation Pathway

The OXPHOS pathway, i.e., the electron transport-linked phosphorylation, is a metabolic pathway responsible for producing chemical energy in the form of adenosine trisphosphate (ATP). This pathway consists of five multi-subunits protein complexes and two mobile electron carriers (coenzyme Q (CoQ) and cytochrome *c*). OXPHOS generates an electrochemical transmembrane gradient through the first four multi-protein complexes (I–IV), which consist the mitochondrial respiratory chain (MRC) that fuels ATP synthesis through the last complex (V) of OXPHOS, the ATP synthase complex [5]. In this pathway, the two electron donors (the reduced nicotinamide dinucleotide (NADH) and the reduced flavin adenine dinucleotide (FADH2)), which are products of the glycolysis, the β-oxidation of fatty acids, and the TCA cycle, start passing along electrons during oxidation to complex I (ubiquinone oxidoreductase-NADH) or complex II (succinate dehydrogenase), respectively, and then to electron carrier coenzyme Q or ubiquinone. Reduced ubiquinol transfers electrons to complex III (ubiquinol: cytochrome *c* oxidoreductase) and, more specifically, to cytochrome *c_1_*. Then, the second soluble electron carrier, cytochrome *c*, transfers the electrons to complex IV (cytochrome *c* oxidase or COX), which catalyzes molecular oxygen (O_2_) to water (H_2_O) at the final stage. The downstream transport of electrons results in a proton gradient as protons are pumped into the intermembrane space, thus providing the energy to ATP synthase (complex V) to convert adenosine diphosphate (ADP) and inorganic phosphate (Pi) into ATP.

While the complexes I, III, IV, and V are under dual genomic control since their structural proteins are encoded from both genomes, complex II’s structural proteins are encoded only by the nDNA. Additionally, numerous ancillary proteins called assembly factors (all being encoded by the nDNA) are necessary for the biogenesis and assembly of the five multi-subunits complexes of OXPHOS.

### 1.3. The Mitochondrial Machinery for Protein Import and Assembly

Although mitochondria have their own DNA, less than 1% (13 structural subunits of OXPHOS) of the roughly 1500 mitochondrial proteins are produced in the matrix. The vast majority of the mitochondrial proteome is encoded by the nDNA (spread across most chromosomes) and expressed as protein precursors (preproteins) in the cytosol. The mitochondrial preproteins either carry a cleavable mitochondrial targeting signal (MTS) peptide in their sequence or various internal targeting signals, which in both cases guide them to the mitochondria. The MTS peptide (predicted by the Mitoprot logistic program: https://ihg.helmholtz-muenchen.de/ihg/mitoprot.html, accessed on 1 November 1996) or leader peptide (L) is typically a 15–50 amino acid (aa) peptide, mostly in the N-terminus of the protein precursors, with some exceptions found in C-terminus [21]. Cytosolic chaperones facilitate the transfer of the preproteins to the general entry gate of mitochondria, the translocase of the outer membrane (TOM) complex. Nearly all mitochondrial preproteins are imported via the TOM complex, and they are subsequently sorted into one of the mitochondrial sub-compartments using the mitochondrial protein import-sorting machinery. Depending on the final destination of mitochondrial proteins, Pfanner’s group [22] suggests five major protein import pathways. The majority of matrix proteins and inner-membrane proteins, which are synthesized with N- (or C-) terminal cleavable MTS peptides (pre-sequences), utilize the classical import pathway (or pre-sequence pathway) consisting of the TOM complex, the translocase of the inner membrane (TIM23), and the pre-sequence translocase-associated motor (PAM). Once these preproteins enter into the matrix, the mitochondrial processing peptidase (MPP) [23] cleaves off the MTS peptides. Pre-proteins with internal targeting signals utilize the other four (4) described pathways, with the final destination defining which of the pathways will be employed. Components of these pathways are the small TIM chaperones of the intermembrane space, the carrier translocase of the intermembrane space, the translocase of the inner membrane (TIM22), the sorting assembly machinery (SAM), the mitochondrial intermembrane space import and assembly (MIA), translocase of the inner membrane (TIM), and the mitochondrial import complex (MIM). Ongoing research will probably reveal additional import routes and complexity given that the pathways responsible for several precursor proteins remain still un-identified.

### 1.4. Mitochondrial Genetic Disorders

While mitochondrial disorders represent a heterogeneous group of rare inherited genetic disorders with varying severity and clinical outcomes, they are generally characterized by respiratory chain dysfunction and abnormal mitochondrial function. The overall estimated prevalence of these disorders is one to three cases per 20,000 individuals [24], with lifelong symptoms that can appear from very early (newborn) to later in life. Unsurprisingly, mitochondrial disorders often affect organs with high metabolic needs such as muscles, heart, liver, kidneys, central nervous system (CNS), peripheral nervous system (PNS), as well sensory organs (eyes and ears) [13,25]. It has been reported that some mitochondrial disorders can affect only a single organ (as in Leber hereditary optic neuropathy (LHON) [26] and in mitochondrial non-syndromic hearing loss and deafness), but often, these disorders cause multi-system dysfunction.

Primary mitochondrial disorders (PMDs) are genetic disorders caused by pathogenic variants in either mitochondrial or nuclear genes coding for mitochondrial respiratory chain and related proteins. The first time that mitochondrial disorder was reported was in 1962 [27] in a patient with severe hypermetabolism, while in 1988, the first mutations in mtDNA were associated with human disease [26,28]. Since then, pathogenic variants in more than 400 genes of both mitochondrial and nuclear origin have been reported in primary mitochondrial disorders.

Mitochondrial dysfunction has also been observed in numerous neurodegenerative disorders, such as Alzheimer’s disease (AD), Parkinson’s disease (PD), and Friedreich’s ataxia (FRDA); in age-related disorders; in metabolic disorders; as well in traumatic brain injury, ischemic stroke, and in a wide spectrum of human cancers [6,15,29].

#### 1.4.1. Mitochondrial Disorders Attributed to Genes Encoded by mtDNA

The mtDNA is highly polymorphic among individuals (~50–60 neutral polymorphisms between two individuals). The lack of protective histones and limited repair mechanisms make the mitochondrial genome highly susceptible to oxidative damage [30,31], with an estimated mutation rate 10–20 times higher than that of nDNA [32]. Mutations in mtDNA include: (a) large-scale rearrangements (single deletions or duplications) that are always heteroplasmic and (b) point mutations (either homo- or heteroplasmic). In the latter, pathogenic point mtDNA mutations frequently coexist with wild-type (w/t) mtDNA (a phenomenon termed heteroplasmy), with higher levels of mutation accumulating in post-mitotic tissues such as skeletal muscle, heart, and the central nervous system [33]. In these cases, individuals develop symptoms/clinical phenotype when the mutated mtDNA exceeds a threshold level (usually 60–90%) compared to normal [34]. Interestingly, family members may have different levels of mutated mtDNA, and even individuals may experience different levels of mutated mtDNA in different organs and tissues [35].
Large-scale rearrangements are associated with:
Sporadic progressive external ophthalmoplegia (PEO) [36];Kearns–Sayre syndrome (KSS) [36];Pearson’s syndrome [37]; Leigh syndrome (LS) (rarely) [38,39]
Homoplasmic point mutations are associated with:
Leber hereditary optic neuropathy (LHON) [26]
Heteroplasmic point mutations are associated with:
Mitochondrial encephalomyopathy with lactic acidosis and stroke-like episodes (MELAS) [40];Myoclonic epilepsy with ragged red fibers (MERRF) [41];Neurogenic weakness ataxia and retinitis pigmentosa (NARP) [42]; Leigh syndrome (LS) [38,39]

#### 1.4.2. Mitochondrial Disorders Attributed to Genes Encoded by nDNA

Mitochondrial disorders can also be caused by mutations in nDNA, specifically in genes that encode either (a) structural proteins of the core subunits of the mitochondrial respiratory chain (complex I–IV) and ATP synthase complex V [43] or (b) no-structural proteins [44]. The second category includes: (i) assembly factors of the respiratory complexes; (ii) proteins responsible for the transcription and translation of mtDNA; (iii) proteins involved in mtDNA replication and maintenance of its integrity and abundance; (iv) proteins necessary for mitochondrial import; (v) proteins involved in iron homeostasis as well as in coenzyme Q10 biogenesis; and (vi) proteins involved in mitochondrial metabolism.

### 1.5. Cytochrome c Oxidase (COX) Deficiency

COX, the terminal enzyme of mitochondrial respiratory chain, is a 14-subunit holoenzyme that catalyzes the electron transfer from reduced cytochrome *c* to molecular oxygen in order to produce water and thus facilitates the generation of the proton gradient across the inner mitochondrial membrane that is used to synthesize ATP [45,46]. This electron transfer occurs through COX’s redox centers, two heme A moieties (*a* and *a*_3_), and two copper centers (Cu_A_ and Cu_B_) [47,48].

Three of the COX’s subunits are encoded by the mtDNA (MT-CO1, MT-CO2, and MT-CO3), and form the catalytic core of the holoenzyme, while the remaining eleven (COX4, COX5A, COX5B, COX6A, COX6B, COX6C, COX7A, COX7B, COX7C, COX8, and NDUFA4/COXFA4 [46,49]) are encoded by nDNA. Interestingly, some of the nuclear encoded subunits of COX (4, 6A, 6B, 7A, 7B, and 8A) appear with different isoforms that are tissue-, developmental-, and species-specific isoforms [50]. The heme *a* and the heme *a*_3_-Cu_B_ binuclear centers are associated with MT-CO1, whereas MT-CO2 contains the Cu_A_ center [51]. Additional, around 30 nDNA-encoded ancillary factors, including copper chaperones, also called COX-assembly proteins, are involved in the biogenesis and assembly of COX holoenzyme. To date, mutations in the subunits encoded by both genomes have been associated with disorder (*MT-CO1* [52], *MT-CO2* [53], *MT-CO3* [54], *COX4I1* [55], *COX4I2* [56], *COX5A* [57], *COX6A1* [56,58], *COX6A2* [56,59], *COX6B1* [60], *COX7A1* [56], *COX7A2* [56], *COX7B* [61], *COX8A* [62], and *NDUFA4* or *COXFA4* [63]), while the majority of isolated COX deficiencies are caused by mutations in COX assembly factors. Specifically, disorder-causing mutations were found in genes such as *SURF1* (the first identified nuclear gene encoding a factor involved in the biogenesis of COX and being mutated in the neurodegenerative Leigh’s syndrome with COX deficiency) [64,65]; *SCO2/SCO1* (involved in mitochondrial copper pathway) [66,67]; *COX10* and *COX15* (involved in heme A biosynthesis) [68,69,70]; *COX14* (or *C12ORF62*; involved in COX assembly) [71]; *COX20* (or *FAM36A*; involved in MT-CO2 stabilization) [72]; *COA3* (*CCDC56* or *MITRAC12;* involved in *MT-CO1* maturation) [73]; *PET100* (involved in COX biogenesis) [74]; *PET117* (involved in the assembly of MT-CO2) [75]; *COA5* (*C2ORF64)* [76]; *COA6* [77]; *COA7* [78]; *COA8* (previously known as *APOPT1*) [79]; *FASTKD2* [80]; *LRPPRC*; and *TACO1* (essential for COX expression) [81,82].

### 1.6. SCO2

The *synthesis of cytochrome c oxidase 2* (*SCO2*) gene is located on chromosome 22 (22q13.33), and it encodes a precursor mitochondrial protein of 266aa. This precursor protein (full length) harbors a mitochondrial targeting sequence (MTS or L) on its N-terminus (1-41aa) that facilitates its transportation to mitochondria and, more specifically, to the inner mitochondria membrane. Sco2 protein acts as assembly factor of COX (complex IV) and is involved in the biogenesis of COX subunit II (MT-CO2), an essential core subunit of complex IV [83]. Sco2 contains a highly conserved copper-binding motif (CXXXC), and acting as metallochaperone, it participates in transport of copper to the Cu(A) site of ΜΤ-CO2 along with Sco1 homolog protein and Coa6, downstream of Cox17 [84,85,86,87,88,89]. In addition, Sco2 acts as a thiol-disulfide oxidoreductase to regulate the redox state of the cysteines in Sco1 [90]. Sco2 protein is also involved in mitochondrial redox signaling [91] and the p53 regulatory pathway in mitochondria [92,93].

Mutations in *SCO2* are often reported in cases of COX deficiency and have been associated with severe phenotypes and different clinical outcomes, such as myopathies, cardiac hypertrophy, neuropathies, and Leigh syndrome [66,94,95,96,97,98,99,100,101,102,103,104,105,106,107,108,109,110,111,112,113,114,115,116,117,118,119,120,121]. Initially, mutations in *SCO2* were found in three unrelated infants with fatal cardioencephalomyopathy and COX deficiency [66]. To date, nearly 100 patients with pathogenic variations in the *SCO2* gene have been reported (Table 1). A recurrent p.Glu140Lys (E140K) mutation has been described in at least one allele in the majority of the patients, raising the possibility of a hot-spot mutation. Patients homozygous for the E140K mutation have a delayed onset of disorder and longer survival compared with patients compound heterozygous for the E140K mutation [122]. Characterization of the E140K mutation in Sco2 protein from our group [84] revealed decreased affinity to copper when compared to the w/t Sco2 protein.

Homozygous deletion of *SCO2* mice (knock-out mice *SCO2^KO/KO^*) is embryonic-lethal [123]. Heterozygous *SCO2* knock-out/knock-in (*SCO2^KO/KI^*) mice carrying a *SCO2* knock-out (KO) allele and a *SCO2* knock-in (KI) allele with the missense mutation E129K (corresponding to the E140K mutation found in almost all human *SCO2*-mutated patients) and homozygous *SCO2* knock-in mice (*SCO2^KI/KI^*) carrying the missense mutation E129K in both alleles have been shown to be viable and fertile despite the muscle weakness and reduction in COX activity [123]. Additionally, heterozygous *SCO2^KO/KI^* mice have increased fat mass associated with reduced β-oxidation and increased adipogenesis markers, reduced insulin receptor beta (IR-β levels in adipose tissue), reduced muscle glucose transporter 4 (Glut4) levels, and an impaired response to the insulin-tolerance test consistent with insulin resistance [124]. Even though these mice models do not recapitulate human disorder, they are still a valuable preclinical tool for testing new therapeutic approaches for COX deficiencies.

## 2. Protein Replacement Therapy for Monogenetic Disorders

Monogenic disorders can be recognized by impressive familial inheritance patterns (dominant/recessive as well paternal/maternal germ line) and are characterized by the abnormal expression or the lack of expression of a crucial protein due to mutations in a respective, single gene [125]. Most mutations are nucleotide substitutions and microdeletions [126]. However, the phenotypic manifestation depends not only on the mutations in this single gene but may depend also on other genetic variations in this locus, epigenetic alterations, post-translational modifications, and/or environmental factors [127]. Based on ESHRE PGD consortium [128], some monogenic disorders are cystic fibrosis, myotonic dystrophy, Huntington’s disease, beta thalassemia, fragile X syndrome, spinal muscular atrophy, hemophilia A, hemophilia B, Duchenne muscular dystrophy, human leukocyte antigen-associated disorders, sickle cell anemia, Friedreich ataxia, and others [129]. As of 2022, unfortunately, there is still no specific treatment for the most of the estimated ~10,000 or more rare disorders [130,131].

Protein replacement therapy (PRT) aims to replace dysfunctional proteins in monogenic disorders, where key proteins are abnormal or not expressed at all. The replacement of these proteins has been proposed using the respective normal or modified proteins generated by high-edge technologies, such as genetic or recombinant protein engineering, exploiting several expression systems (mammalian cells, bacteria, yeast, phage display, etc.). Therapeutic proteins can be manufactured in a high-yielded and reasonable, cost-effective manner and totally purified, being biologically active, folded, and carrying all the desired post-translational modifications. Furthermore, since PRT with therapeutic proteins does not require the delivery of genetic material into the recipient cell nucleus, the most important issues of gene therapy (i.e., insertional mutagenesis) are eliminated.

To date, the use of PRT is focused on treating rare disorders with missing or deficient proteins. In this case, the therapy falls into the category of “orphan drugs”, which are referred to treat disorders affecting 200,000 or fewer individuals in the USA and fewer than 250,000 individuals in Europe [132].

The first therapeutic protein ever used was insulin for the treatment of diabetes mellitus. In August of 1921, Nicolae Paulescu presented for first time the effect of pancreine (later called insulin), which was extracted from animal pancreas, on diabetes symptoms [133]. Also in 1921, the Canadians Banting and Best managed to isolate amounts of (impure) insulin from the islets of Langerhans. In 1922, insulin from a bovine pancreas was administered to a diabetic person, aged 14, who avoided death from a diabetic coma [134], garnering a Nobel prize for Banting and Macleod. The pharmaceutical company Eli Lilly (1923) offered to produce large quantities of higher-purity insulin and, in 1982, achieved the first production of biosynthetic human insulin using recombinant DNA techniques.

The first PRTs approved as “orphan drugs” were blood factors for hemophilias as well recombinant enzymes, referred to enzyme replacement therapies (ERTs) [135], for lysosomal storage disorders (LSDs) [136]. In the past few decades, therapeutic approaches were developed for hemophilia A or B, with the administration of recombinant factor VIII or factor IX, respectively [137]. Furthermore, there are eleven (11) PRTs approved by the Food and Drug Administration (FDA) or European Medicines Agency (EMA) for a number of rare monogenic LSD disorders such as Gaucher disease, Fabry disease, and mucopolysaccharidosis type I [138].

The main limitation for these therapies is the issues raised on the effectiveness of the administration route (oral, i.v., i.a., i.m.) regarding metabolism and clearance before the delivery to the targeted tissue. Furthermore, another main issue refers to the effective delivery into the CNS and the crossing of the blood–brain barrier (BBB) due to the large size of the recombinant proteins. While safety is not an issue for PRT, hypersensitivity may occur, leading to allergic reactions and the production of antibodies to the respective exogenous protein, leading also to neutralization/loss of its function. Finally, the manufacturing and purification cost stands as an obstacle for this therapeutic approach [130].

### 2.1. Viral-Mediated Gene Transfer and Expression as a Therapeutic Approach for the Monogenic Disorders

Gene therapy aims to treat several genetic disorders through the alteration of the patient’s genetic profile by delivering the functional therapeutic gene into the host cell nucleus. Viral-mediated gene therapy involves (a) the expression of a protein using genetically engineered viruses or (b) the suppression of a pathogenic gene (by using RNA interference technology (RNAi) or zinc-finger nucleases or CRISPR-Cas9-based approaches) to inhibit gene expression. Regarding the genetic engineering of the virus, it can take place by injecting the recombinant vector (with the therapeutic gene) directly into the target tissue, i.e., the organ, or the cells can be ex vivo genetically modified and re-injected into the patient in the context of cell therapy (autologous or allogenic). In the case of viral-mediated gene transfer by using engineered retroviruses, lentiviruses, or adeno-associated viruses (AAVs), the therapeutic gene replaces part of the viral genome delivered into the nucleus, and the corresponding protein is expressed [139,140].

For the construction of an AAV vector, a therapeutic gene, a promoter, and regulatory elements up to 5 kb in size are inserted into the AAV virus (single-stranded DNA genome of approximately 4.8 kb). Directing the tropism of the AAV vector to target specific cells and tissues relies on the capsid proteins, which can be engineered to develop the desired tropism [141]. AAVs for gene therapy are modified not to encode the viral proteins needed for the integration into the human genome. To avoid the loss of the AAV vector, non-dividing cells are preferred.

Retroviruses (single-stranded RNA genome of between 7 to 12 kb) use reverse transcription to permanently integrate into the human genome independently from the division state of the cell. Lentiviruses, which are included in the retroviruses family, have been efficiently exploited as a platform for ex vivo gene therapy using hematopoietic stem cells (HSCs) [142].

In the early 1990s, the excitement surrounding gene therapy was so great that treating human monogenic disorders seemed within reach. However, the death of one patient, 17-year-old J.G., after transfection with an adenoviral vector (different from the one used nowadays) encoding the *OTC* gene to treat the ornithine transcarbamylase deficiency (OTCD) [143] he suffered from was enough to stop any enthusiasm around the idea of gene therapy. In an instant, the field of gene therapy collapsed, taking with it the grandiose promises of miracle cures for more than a decade. However, the field eventually recovered. In the 30 years since J.G.’s death, private and public companies have invested billions of dollars in their efforts to cure disorders by changing or replacing deficient genes. Nowadays, AAV-modified vectors are used safely in more than 200 human clinical trials [130]. There are three (3) AAV-based gene therapies approved for monogenic disorders, with the first one targeting lipoprotein lipase deficiency [144] (withdrawn in 2017 due to commercial issues), followed by the approval in 2017 of a treatment for RPE65 mutation-associated inherited retinal dystrophy, followed by the approval in 2019 of onasemnogene abeparvovec (ZOLGENSMA^®^) by replacing the missing or nonworking *SMN1* gene with a new and working *SMN1* gene for the treatment of spinal muscular atrophy (SMA). Gene therapy for β-thalassemia requires the transfer of the gene into HSCs using viral vectors that direct the regulated expression of *β-globin* to therapeutic levels [145]. For example, a gene therapy approach via a lentiviral vector that carries the gene for human β-globin recently received marketing authorization for betibeglogene autotemcel (beti-cel) (ZYNTEGLO^®^) for the treatment of transfusion-dependent thalassemia (TDT) and is one of the most expensive therapies to be marketed in the U.S.A. EMA’s Conditional Marketing Authorization (CMA) for ZYNTEGLO^®^ was finally suspended due to the extremely high cost.

Additional efforts are underway, such as gene therapy mediated by retro- or lenti-viral vectors (currently in phase III clinical trials (NCT02906202) and a long-term follow-up study (NCT02633943)) or through gene editing [146] in hematopoietic stem and progenitor cells (HSPCs) or in induced pluripotent stem cells (iPSCs). HSPCs can be modified directly, or alternatively, somatic cells (e.g., fibroblasts) can be first reprogrammed into iPSCs before being modified. In both cases, the genetically modified HSPCs are re-injected back into the patient. On the other hand, direct in vivo transfection of HSPCs includes the risk of not every cell being modified.

The latest technologies aim to correct β-thalassemia mutations either by introducing a normal globin gene (β- or γ-), by suppressing the expression of another gene (gene knockdown) such as α-globin [147,148], or through gene editing by using nucleases [149,150,151,152]. However, retroviral vectors are known to be relatively mutagenic, and risk management is required; thus, there is a need to explore alternative genetic and cellular therapeutic approaches.

Concerning the treatment of mitochondrial disorders, which are also characterized as rare disorders, with 90% of them caused by a mutation in a single nuclear gene, it is limited to supporting care strategies with symptomatic, pharmacological, gene, and protein therapy [153]. Because most of the mitochondrial proteins are encoded by nuclear genes [154], and most mitochondrial disorders have clinical features consistent with multiorgan disorder, a successful gene therapy should include a vector that is delivered systemically and must be able to cross the BBB. Furthermore, gene therapy for mitochondrial disorders arising from nDNA mutations faces the same obstacles as gene therapy efforts, but with respect to mitochondrial DNA mutations, additional complexities are raised due to the heteroplasmy phenomenon, the necessity to target mitochondria exclusively, the localization that the gene products can have in the different mitochondrial compartments, and due to the differences of the mitochondrial vs. the universal code [155].

Gene therapy has offered some hope for treating Leber hereditary optic neuropathy (LHON) disease, which has been shown to be partly related to COX deficiency. In a mouse model of LHON derived from a mutation in the mitochondrial *ND4* gene [156], disorder reversal was observed by optimizing the ectopic expression of an AAV virus in the nucleus, containing a modified version of *ND4* for mitochondrial import and translation [157]. LHON disease is particularly suitable for this approach, as the ganglion layer of the retina (which because of its degeneration is decisive for vision loss) can be easily approached. Gene therapy as a means of treating patients with LHON is still in its infancy, with trials underway in China and the USA as well as for Friedreich ataxia (Table 2). Unfortunately, this approach is hampered by accessibility issues for most COX-related disorders that affect other organs or tissues.

Furthermore, efforts have been conducted in a pre-clinical stage also using AAV-mediated gene transfer for Barth syndrome; Friedreich ataxia; *NDUFS4-*, *NDUFS3-*, and *SURF1*-related Leigh syndrome; ethylmalonic encephalomyopathy; mitochondrial neurogastrointestinal encephalomyopathy (MNGIE); MPV17 deficiency; TK2 deficiency; and SLC25A46-related neuropathy [158].

Despite the clear evidence of clinical efficacy of AAV vectors, another drawback of the technology is the possibility of generation of neutralizing antibodies against the AAV vectors. Alternatively, retroviral transfections “hide” the danger of the “off-target mutagenicity” due to their integrative ability into the genome, resulting in possible deregulation of oncogenes and tumorigenic transformation [159]. The complex and cost-effective manufacturing procedure is also a limitation for viral gene therapies as well. Overall, the early stage of development of gene and cell therapies requires more research to ensure safety and consistency.

### 2.2. Non-Viral Systems for Delivery Gene Sequences

The use of viral systems is an attractive and very effective method of restoring deficient genes, with high levels of transfectability. However, there are several risks associated with the use of this technology, with the main ones being immunogenicity, ectopic chromosomal integration of viral DNA into oncogenes or tumor suppressors (insertional mutagenesis), and cytotoxicity. Some non-viral systems are used to deliver nucleic acids as gene transfer platforms and include the delivery of small DNA (oligo-DNA), large DNA (plasmid DNA, pDNA), and RNA (ribozymes, siRNA, mRNA) [139]. Strategies regarding nucleic acid delivery are divided in physical and particle-mediated methods. Physical methods exploit transient and mechanical disruption of the membrane to introduce the nucleic acid into the cells. Among physical methods are microneedle and jet injection, gene gun, electroporation, and sonoporation. Particle-mediated methods consist of metallic particles, cationic polymers, cationic lipids, PTDs, virus-derived particles, and exosomes [160].

The nucleic acid technology with the greatest potential nowadays is that of in vitro transcribed (IVT) mRNA, which can potentially be applied to various fields, including vaccines, gene editing, cell reprogramming applications, cancer immunotherapy, and of course PRT, including the de novo synthesis of a deficient or non-functional protein in target cells for a wide range of diseases [161]. Clinical trials are conducted with vascular endothelial growth factor (VEGF, NCT03370887) and the CFTR (NCT03375047) [162]. Encouraging results have also been reported for PRT using mRNA as the therapeutic intervention for myocardial infarction and heart failure (NCT03370887) [163] and for genetic lung disorders [164]. There are also pre-clinical studies for hemophilia A (factor VIII deficiency) and hemophilia B (factor IX deficiency) [165]. However, for IVT-mRNA-mediated therapy to be considered as a long-term PRT of monogenic disorders, repeated administrations of the IVT-mRNA should be applied [166].

The main advantage of non-viral systems for PRT, such as the IVT-mRNA technology, is their safety feature. IVT-mRNAs do not integrate into the host genome, so they do not trigger oncogenic mutations, in contrast to gene therapy via viral vectors or DNA-based methods. IVT-mRNA is degraded after few days in the cytoplasm by the normal mechanisms of the cell itself [167]. Therefore, neither the deactivation nor the removal of IVT-mRNA is needed compared to viral vectors, as in the case of unexpected severe toxicity with either the use of “safety switches” or “suicide genes” (suicide genes) [167]. An additional advantage of IVT-mRNA over other non-DNA integration methods, such as episomal DNA plasmids, is that IVT-mRNA-mediated protein expression is carried out independently of division stage. pDNA enters the nucleus preferentially during the mitotic phase, resulting in the most efficient protein expression being carried out in dividing cells. Furthermore, comparing the IVT-mRNAs with the recombinant therapeutic proteins, the main advantage is the lack of the complex purification procedures. In addition, good manufacturing practice (GMP) costs are up to 10 times lower [168].

## 3. Protein Transduction Domains (PTDs) or Cell-Penetrating Peptides (CPPs)

Protein transduction domains (PTDs), also known as cell-penetrating peptides (CPPs), membrane transduction peptides (MTPs), and Trojan peptides, are short peptide sequences of 5-30 aa that have the ability to cross cell membrane barriers by energy-dependent or independent mechanisms [169]. These peptides can be attached (covalently or non-covalently) [170,171] to a number of bio-active cargos [172], such as peptides and proteins; small-molecule drugs (Doxorubicin, Paclitaxel); nucleic acids (pDNA, siRNA, miRNA); nanoparticles (liposomes, micelles, carbon nanotubes (CNTs), gold nanoparticles (GNPs)); or fluorophores (FITC, rhodamine, Cy5), which can be transduced intracellularly.

The first report of a protein passing through cell membranes of cultured cells came from two independent research groups in 1988 [173,174]. While they were studying human immunodeficiency virus type 1 (HIV-1), they discovered that purified or chemically synthesized HIV-1 TAT (trans-activator of transcription) protein was readily transferred intracellularly when simply added to the culture medium of cells without any transfection reagents and localized to the nucleus. Further efforts indicated the protein region that is necessary for cellular uptake [175,176] and later the short basic domain of the HIV-TAT protein, an 11aa peptide (YGRKKRRQRRR, residues 47–57) enriched in arginine (R) and lysine (K) residues [177]. This discovery provided initial proof of the feasibility of intracellular delivery of proteins and spurred further studies and reports of more PTDs that could direct a variety of cargos into cells both in vitro and in vivo. The second PTD/CPP that was reported few years later, which since then is also often used, is the Penetratin or Antennapedia (Antp) peptide derived from a Drosophila homeotic transcription factor [178,179]. The first PTD in vivo delivery in an animal model while carrying cargo many times its size in a functional form was reported from Dowdy’s group in 1999 [180]. In that report, the authors were able to deliver biologically active 120 kilodalton (KDa) beta-galactosidase (β-GAL) protein to all mouse tissues (including the brain) after intraperitoneal injection (i.p.) by fusing the β-GAL to the TAT (11aa) peptide from the HIV-1 TAT protein. This in vivo methodology highlighted the potential for new therapeutic approaches and opened new possibilities for the development of vaccines and protein therapies for cancer and infectious diseases. Up to now, around 1700 unique PTDs/CPPs have been reported [172] for both in vitro and in vivo applications. Advantages of PTD technology include high internalization, ease of synthesis, potential for sequence modification, low cytotoxicity, low micromolar concentrations in vivo and in vitro, no need to use any chiral receptors, and no significant membrane damage.

PTDs/CPPs can be classified into different groups based on their characteristics. They can be either linear (most of them) or cyclic. Based on their origin, PTDs derived from proteins with the ability to penetrate cell membranes are considered natural products (i.e., TAT and Penetratin), whereas if chemically synthesized, they are synthetic products (i.e., the polyarginine family, Rn 6 < n < 12) [181]. PTDs can also be chimeric, such as the 27 aa long peptide galparan (as well the peptide Transportan, a galparan substituted with lysin in position 13), containing the first thirteen (13) aa from the amino terminus of the neuropeptide galanin and the fourteen (14) aa long wasp venom peptide toxin, mastoparan, in the carboxyl terminus connected via a lysine [182]. Another way to group PTDs is based on their sequence (physical and chemical properties), creating three categories [183]:

Cationic peptides that contain a high relative abundance of positively charged amino acids, such as lysine (K) and/or arginine (R) (i.e., TAT [177] and polyarginines [181]);

Amphipathic peptides, usually chimeric or fused peptides, having sequences that contain both polar (hydrophilic)-charged amino acids and non-polar (hydrophobic) amino acids and both positively and negatively charged amino acid sequences (i.e., MPG peptide (HIV glycoprotein 41/SV40 T antigen NLS)) [184]; 

Hydrophobic peptides, having only non-polar residues with low net charge or hydrophobic amino acid groups (i.e., Pep-7 peptide) [185].

The mechanism by which PTDs enter the cell is not totally clear and can vary based on the nature and type of PTD (nature, charge, sequence, length, conformation), the experimental set up (i.e., the cell types), and the fused cargo or drug [186,187]. Two mainly different pathways have been proposed to explain cell penetrance: direct penetration and endocytosis [188]. Direct penetration is an energy- and receptor-independent pathway that has been proposed for PTDs associated with small cargos. The process has been described using three models: 

The “carpet-like” model or “membrane thinning” model, where the PTDs or PTDs/cargoes are transported into the cell by charge interaction between negatively charged phospholipid and cationic PTDs, resulting in carpeting and thinning of the membrane, respectively. When the concentration of PTD is above a threshold, the PTD is transported into the cell;

The “pore formation” or “barrel-stave” model, where PTDs can enter through the plasma membrane either by toroidal pores or barrel-stave pores [183]. Thereafter, PTDs are delivered into the cytoplasm;

The “inverted-micelle” model, where the hydrophobic residues of the PTD interact with the lipids of the cell membrane to form micelles. These micelles are internalized and release the peptide into the cytosol through inversion of the micelles. 

Endocytosis is an energy-dependent, active transport pathway mainly facilitated by phagocytosis (usually for large molecules) and pinocytosis. The latter can be further divided into macropinocytosis, caveolin-mediated endocytosis, clathrin-mediated endocytosis, and clathrin- and/or caveolin-independent endocytosis.

## 4. Deliverable Recombinant Protein Delivery Mediated by PTD Technology

The cell membrane creates a physical barrier separating the outside from the inside of the cell and actively participates in the transport of biomolecules between these two spaces. PTDs is a promising technology that facilitates the transport of proteins through cell membranes, thus allowing us to consider a protein-based therapeutic modality to treat human disorders that traditionally have been avoided due to membrane permeability issues. Since their discovery [173,174,178,179], the number of known PTDs is steadily growing, and their transduction efficiency continues to improve [189,190]. In vitro and in vivo models of neurological disorders, rare genetic disorders, heart diseases, and cancer are some models in which PTDs have been evaluated in the frame of therapeutic approaches over the past 20 years [191,192].

Furthermore, by using leader peptide sequences, PTDs can also be used for delivering their cargo proteins to a specific subcellular compartment, such as mitochondria [193,194]. For example, a cell-penetrating artificial mitochondria-targeting peptide (CAMP) conjugated with human metalloprotein 1A (hMT1A) restored mitochondrial activity and ROS production in in vitro PD model, while injection of CAMP-hMT1A fusion protein in the brain of a mouse model of PD rescued movement impairment and dopaminergic neuronal degeneration [195]. FRDA is a disease caused by the silencing of the *Frataxin* (*FXN*) gene and reduced levels of the nuclear-encoded mitochondrial protein that functions primarily in iron-sulfur cluster synthesis [196]. Fusion of TAT peptide to full-length FXN protein [197] or to mature FXN protein harboring another MTS signal [198] drove the fusion proteins to the mitochondria of cells derived from FRDA patients as well as in FXN-deficient neurons and increased cell survival, decreased neuritis degeneration, and reduced apoptotic markers [197,198]. Animal studies using FRDA models [199] revealed that fusion FXN protein restored the activity of the succinate dehydrogenase and significantly increased lifespan [197,198]. In a Parkinson’s disease mouse model, the same fusion protein (TAT-FXN), delivered by i.p. injection, crossed the BBB and protected the dopaminergic neurons [200]. In another study with a mouse model of cerebral ischemia, Cao and co-workers introduced TAT-HA-Bcl-xL fusion protein by i.p. injections and observed robust protein transduction in neurons in various brain regions and decreased cerebral infarction in a dose-dependent manner [201].

While this new therapeutic approach is the focus of extensive research, including multiple in vitro and in vivo applications, no PTD modalities have been approved yet by the FDA. Key challenges that still need to be addressed before clinical application include the following:

Stability efficacy: Although most PTDs penetrate readily into most organs, they show rapid clearance from the circulation. Moreover, it is mostly unclear if the entry mechanisms are the same across different cell types or across different PTDs. Thus, any effort to improve stability or penetration efficiency would require understanding of these underlying molecular pathways.

Toxicity: Data from in vitro studies are very encouraging. Cationic PTDs, such as TAT and Penetratin, are less toxic, and cells can tolerate much higher concentrations of these PTDs than amphipathic PTDs, such as Transportan. Especially, no toxicity of TAT and Penetratin was observed in cells in the usually working concentration when assays such as cell viability proliferation and leakage of lactate dehydrogenase (LDH) were performed. Moreover, preclinical studies are promising, but they are limited and without long exposure time of PTDs cargos in the animal models. Still, the overall effect of PTDs on cell/organism/organ function remains to be evaluated.

Lack of tissue/organelle specificity: Most PTDs do not target specific tissues or organelles. Global accumulation of cargo proteins can lead to unwanted toxicities and organ damage. Efforts to supplement these transduction domains with additional directing sequences are ongoing, but the effectiveness of this approach remains unproven.

However, a number of cell-specific PTDs can be identified [202] mainly via the use of peptide phage display libraries [191,192].

### PTD-Mediated PRT for Mitochondrial Disorders: Transduction of the Human Recombinant TAT-Sco2 Fusion Mitochondrial Protein

The lack of effective treatment for mitochondrial disorders combined with their genetic complexity as well as the risks involved in gene therapy led to the imperative need of alternative treatments. As previously mentioned, PTDs allow proteins to enter cells, while with appropriate modifications, these proteins can subsequently enter subcellular target organelles. This technology allows the development of PRT for enzymes and proteins that show reduced or total loss of activity in mitochondria.

The use of PTD-fused proteins targeting mitochondria in the first published PRT approach by using the recombinant human protein in fusion with the TAT sequence (such as the PTD prototype) was developed for lipoamide dehydrogenase deficiency (LAD) by Lorberboum-Galski’s group [203]. Lipoamide dehydrogenase is the E3 subunit of the α-ketoacid dehydrogenase complexes and is involved in carbohydrates and amino acids metabolism. Initially, there were positive in vitro results when using the TAT-LAD fusion protein in LAD-deficient patient cells, where the protein was successfully transduced into mitochondria, processed normally by the MPP and the LAD, and mitochondrial multienzyme pyruvate dehydrogenase complex (PDHC) activity was restored [203]. Then, the study was extended to an in vivo model using LAD-deficient mice. After intravenous administration of the TAT-LAD fusion protein, a time-dependent decrease in its concentration in the blood was observed with a parallel increase in the tissues of interest (liver, heart, and brain). Subsequent analysis of these tissues revealed an increase in LAD activity, particularly in the heart. In mice, where only LAD was administered as a control (without the TAT peptide), no increase in activity was observed [204].

More recently, the same research group developed a PRT for complex I deficiency of the respiratory chain due to mutation of a complex I assembly factor located on chromosome 6 open reading frame 66 (c6ORF66) of nDNA. By using, again, biotechnological methods, the corresponding TAT-open reading frame (ORF) fusion protein (i.e., the TAT peptide fused to the assembly factor of interest) was produced. Its action was then studied in patients’ fibroblasts. Incubation of the recombinant protein with isolated mitochondria from patient cells showed that the TAT-ORF had the potential to enter mitochondria, as the first 34 amino acids of the normal protein have the role of MTS and can be processed by the MPP. Then, incubation of the protein with the same cells showed that the TAT-ORF could enter the cells and even translocate into the mitochondria. Finally, in order to demonstrate the functionality of the recombinant protein, the activity of complex I was studied after incubation of both isolated mitochondria and whole cells with the TAT-ORF compared to the activity of complex I in isolated mitochondria and patient cells that were not incubated or incubated with the ORF alone. The results showed particularly positive results. It is noteworthy that the use of TAT-ORF did not present problems in cell viability. On the contrary, TAT-ORF contributed to an increase of the viability and, in addition, showed a specific effect only in cases where there was a lack of the specific endogenous assembly factor [205].

The first report of the use of PTD technology for the delivery of the human recombinant Sco2 mitochondrial protein as a PRT approach for the disorder, which is caused by mutations in the *SCO2* gene and leads to fatal infantile cardiomyopathy and COX deficiency, was in 2010 by Foltopoulou et al. [193,206]. More specifically, the biotechnologically engineered recombinant 10xHis-Xa_SITE_-TAT-L-Sco2-HA fusion protein (referred as fusion L-Sco2 protein) was produced and purified from bacterial inclusion bodies, which were enriched in the fusion protein and solubilized in 1M L-Arg solution. In particular, the fusion L-Sco2 protein included the full-length normal human L-Sco2 protein, which carried its own MTS (L) and was able to be cleaved by the MPP in fusion with the TAT sequence, such as the PTD. Successful in vitro intracellular delivery of the fusion L-Sco2 protein was followed by its targeting into the mitochondria in the various cell lines (U87, T24, K562) studied in a time-dependent kinetic model. Further experiments using [^35^S]Methionine-labeled recombinant fusion L-Sco2 protein, followed by incubation with isolated intact mitochondria derived from K562 cells, showed that the protein has the potential to mature, presumably via the mitochondrial matrix-located MPP, by recognizing TAT-MTS, which was carefully designed via the cloning procedure, as a new MTS (predicted by the Mitoprot logistic program). Following cleavage by the MPP, the released mature Sco2 protein would be incorporated into the inner mitochondrial membrane. Finally, incubation of fibroblasts derived from a *SCO2*/COX-deficient patient as a cell type model with the fusion L-Sco2 protein led to an increase in COX enzymatic activity (as shown in Figure 1), while the fusion L-Sco2 protein was also identified in the corresponding mitochondrial fractions [193,206]. Moreover, Papadopoulou’s group proceeded to the in vivo biodistribution study of fusion L-Sco2 protein in humanized, healthy Balb/c mice [207]. Initially, the human recombinant fusion L-Sco2 protein was radiolabeled using metastable technetium 99 (^99m^Tc) with successful protein labeling. The stability of [^99m^Tc]Tc-TAT-L-Sco2 was also evaluated, indicating increased levels of stability for up to 24 h following the labeling reaction. The protein-bound radioactivity (% of the total incubated radioactivity) in the plasma was found not to be altered for the 1, 4, and 21 h, supporting remarkable stability of [^99m^Tc]Tc-TAT-L-Sco2 in plasma under in vitro conditions. The [^99m^Tc]Tc-TAT-L-Sco2 protein was administrated in mice through i.v. administration and exhibited fast elimination from the circulation. The highest rates of radioactivity per gram of tissue were found in the liver, followed by the spleen and kidneys. The exogenously administered fusion L-Sco2 protein was detected in the mitochondrial extracts from all tissues examined by Western blotting compared to control extracts. Thus, the in vivo studies showed that the radiolabeled [^99m^Tc]Tc-TAT-L-Sco2 fusion protein is biodistributed in the peripheral tissues and successfully delivered into the mitochondria of all tissues being examined. However, issues concerning the high-purification process of the recombinant fusion L-Sco2 protein and the cost for its GMP production should be carefully considered for its development as a potential therapeutic protein for the treatment of mitochondrial disorders due to *SCO2* mutations.

## 5. IVT-mRNA Delivery Mediated by PTD Technology

PTDs are efficient carriers for the intracellular transduction of various therapeutic cargoes, such as recombinant proteins as well nucleic acids and in vitro transcribed mRNAs (IVT-mRNA). A brief description of the technology of IVT-mRNA therapeutics is described in the following section.

### 5.1. Delivery of IVT-mRNA for Producing Proteins of Given Interest

In the last decade, and especially after the outbreak of the SARS-CoV-2 pandemic, several research groups have demonstrated the tremendous potential and the feasibility of using the IVT-mRNAs, with surprising results regarding the efficiency of transfection and the duration of protein expression but, above all, the safety that accompanies this technology. The first use of IVT-mRNA, which potentially encodes a therapeutic protein and can be intracellularly transported in vivo, was performed by Wolff et al. in 1990, marking the first direct injection of naked IVT-mRNA [208]. The first clinical studies related to this technology were implemented by Gilboa’s group that, in 1996, proposed the potential use of vaccines in the form of transfected—with IVT-mRNA—dendritic cells (DCs) (RNA-pulsed DCs) in patients with very small, possibly microscopic tumors [209]. Towards the end of the 1990s, Hoerr I. and colleagues discovered that by intradermal injection of IVT-mRNA, skin cells expressed the corresponding proteins encoded by the IVT-mRNA. However, most researchers turned to the development of gene therapy using DNA molecules/viral vectors, which ultimately encountered formidable obstacles, and the IVT-mRNA’s therapeutic potential was not fully exploited.

In fact, at the beginning of the millennium, there were very few “allies” of the IVT-mRNA technology who believed that it could indeed be used as a therapeutic molecule. Among the reasons for the mistrust of the IVT-mRNA technology was that the IVT-mRNA is prone to degradation by the harmful enzymes that break it down, the so-called RNases (hence the storage at freezing and super-freezing temperatures). Thus, the IVT-mRNA is a very delicate molecule regarding survival—and potential harm—of the human body. However, Hoerr’s group found that IVT-mRNAs were indeed quite stable in the absence of RNases [210]. In addition, Karikó K. et al. [211] published an exceptional study in 2005 indicating the suppression of the immunogenic reaction caused by the in vivo transfection of the exogenous IVT-mRNA via the optimization of the coding sequence (CDS) by incorporating nucleoside modifications.

Nowadays, through the optimization of the structure and chemical modifications as well as delivery technologies, the IVT-mRNA is in the spotlight, as it is being evaluated in pre-clinical and clinical studies to treat a wide variety of disorders. All this excitement around IVT-mRNA has prompted a wave of biotech companies to capitalize on this technology, now garnering a considerable investment, and an explosion of innovative approaches is expected in the next decade. The applications of the IVT-mRNA technology refer to various fields, including vaccines, PRT to treat monogenic disorders, gene editing, cellular reprogramming, as well cancer immunotherapy. Overall, the market for IVT-mRNA therapeutics was valued at USD 39.90 billion in 2021 and will expand at a compound annual growth rate of 1.7% from 2022–2030 [212]. To date, there are 469 ongoing clinical trials utilizing IVT-mRNA, 58 of which are being tested in children under 17 years of age.

Regarding rare genetic metabolic disorders, several IVT-mRNA approaches have been developed, with very encouraging results. The main idea is the use of lipid nanoparticles for the intracellular delivery of therapeutic IVT-mRNAs into hepatocytes of patients with acute intermittent porphyria (AIP), methylmalonic acidemia (MMA), and mitochondrial OTC deficiency. Currently, Translate Bio (Lexington, MA, USA), PhaseRx (Seattle, WA, USA), and CureVac (Tübingen, Germany) with Arcturus Therapeutics are developing an IVT-mRNA approach for the treatment of OTC deficiency, which is in pre-clinical and clinical stage. Furthermore, Moderna Therapeutics (Cambridge, MA, USA) is developing this approach for MMA, entering phase II clinical trials, and for AIP and propionic acidemia in a pre-clinical stage [213].

### 5.2. IVT-mRNA Synthesis and Optimization Strategies

During the in vitro synthesis of mRNA, a linear plasmid DNA or even a PCR product is the template in a cell-free system. Then, the produced IVT-mRNA, after entering the cytoplasm (via a variety of delivery methods, which will be discussed at length below), allows the cell’s endogenous machinery to proceed in translation of the properly folded and highly functional protein.

The DNA template must be fully linear, and the in vitro transcription reaction is mediated by a T7-, T3-, or SP6-RNA polymerase. The IVT product must carry certain features to serve its purpose: an ORF encoding the corresponding protein, untranslated regions (UTRs), 5′-caps, and a 3′-poly-adenylate [poly-(A)] tail. After transcription, the DNA template is digested with DNase to terminate the reaction, while the subsequent purification of IVT-mRNA may typically consist of enrichment, precipitation, extraction, and chromatography steps. In some therapeutic applications, IVT-mRNA can directly transfect patient cells ex vivo, with subsequent re-injection of the modified cells back into the patient. For direct in vivo delivery of the naked IVT-mRNA to the target site, local delivery is usually performed, but for indirect systemic delivery of IVT-mRNA, a carrier-mediated delivery (perhaps including a targeting moiety) is recommended. Recent advances in nanotechnology and materials sciences have yielded many promising delivery systems, effectively facilitating targeted in vivo delivery.

An important issue for the therapeutic application of IVT-mRNA is its immunostimulatory effect. Exogenous mRNA often exhibits strong immunogenicity through recognition by toll-like receptors (TLRs) [214]. In addition, the secondary structure of exogenous IVT-mRNA itself causes the activation of IFN-inducible protein kinase R and 2′-5′-oligoadenylate synthetase, suppressing translation. In general, RNA has a distinct immunostimulatory pattern, but this can be partly controlled by several modifications of the characteristics of the mRNA molecule itself or of the corresponding carrier used to deliver it [214].

Furthermore, IVT-mRNA is degraded in the cytoplasm relatively quickly, resulting in protein synthesis lasting from a few hours to several days. For this reason, repeated infusions of IVT-mRNAs therapeutic molecules must follow to stabilize protein levels. The pharmacokinetics of IVT-mRNA is determined by its half-life and that of the produced mature protein, resulting from post-translational modifications [215]. The two main factors affecting the bioavailability of exogenous IVT-mRNA in the cytosol are (i) its rapid degradation mediated by “damaging” RNases and (ii) the absence of passive diffusion through the cell membrane due to its high molecular weight and the electrostatic repulsion between the negatively charged IVT-mRNA molecules and the negative charges of the cell membrane, which is coated with proteoglycans [216].

Many strategies have sought to address all these challenges of IVT-mRNA in terms of its stability and immunogenicity. These strategies include the construction of the template with all the necessary modifications to enhance IVT-mRNA stability (expansion of half-life) and subsequent translation, increasing the corresponding protein expression levels. Modifications used to achieve this purpose include poly-(A) tail elongation, 5′-cap modification, modification of UTRs, sequence optimization, and incorporation of modified nucleotides [217].

A mature eukaryotic mRNA during transcription at its 5′-end acquires a cap structure, designated as m7GpppN or m7Gp3N (m: methyl-group, N: any nucleotide) [215,218]. The cap is extensively involved in many cellular processes, including successful translation, splicing, and degradation. Initially, the cap protects the mRNA from rapid degradation by exonucleases, and it is involved in mRNA’s recognition by the eukaryotic translation initiation factor 4E [219]. The cap plays a further role in preventing the activation of innate immune sensors by mRNA recognition [220]. Anti-reverse cap analogs (ARCAs), with a 3’-O-Me-m7G(5)ppp(5)G structure, replace normal caps during transcription of IVT-mRNA to prevent the phenomenon half-caps be oriented in reverse orientation, which makes them unrecognizable by the protective cap-binding proteins [221,222].

In eukaryotic cells, an important mRNA degradation signal is the adenylate-uridylate-rich elements (AREs) in 3′-UTRs [223], which regulate the mRNA exit from the nucleus and the translation efficiency [224] while also orchestrating the subcellular localization of mRNA [225] and its stability in general [226]. The stability of mRNAs is increased, however, when the AREs are replaced with the 3′-UTR of mRNAs, with increased half-life. In various clinical studies and in basic research, the most used 3′-UTRs are derived from *α-* or *β-globin* mRNA. It has been reported that the optimal result is achieved by using two (2) 3′-UTRs of β-globin in a series aligned (in conformation) from head to tail [215]. Furthermore, 5′-UTRs found in numerous orthopox-viruses’ mRNAs have been shown to inhibit degradation by 3′-5′ exonucleases [227]. In addition, the presence of a strong Kozak translation initiation sequence in the 5′-UTR of the mRNA increases translational capacity [228]. The optimal Kozak sequence for eukaryotic mRNAs is suggested to be RCCAUGG, where R is a purine (A or G) [229].

Polyadenylation, i.e., the addition of adenine residues to the 3′-UTR of mRNA, is catalyzed by poly-(A) polymerase, part of the natural mRNA maturation system in eukaryotic cells. The poly-(A) tail acts as a binding site for the poly-(A)-binding protein (PABP), which helps the mRNA exit from the nucleus into the cytoplasm safely, where it binds to subsequent proteins to facilitate translation [230]. The poly-(A) tail turns the mRNA into a very stable molecule since its removal enhances its degradation [231]. The poly-(A) tail in synthetic IVT-mRNA prefers to be elongated up to 100 adenine residues although studies suggest that increasing the number of bases beyond 120 does not further enhance protein expression [232].

The involvement of modified nucleosides during in vitro mRNA synthesis, such as 5-methylcytidine (5mC) and pseudo-uridine (Ψ) instead of cytidine (C) and uridine (U), has been demonstrated to it contribute to reducing the immunogenicity and increasing the stability of IVT-mRNA. This significant optimization reduces cell death and toxicity caused by synthetic IVT-mRNA [233]. The modifications to achieve increased stability and reduced immunogenicity of IVT-mRNA, in addition of Ψ, also include the use of natural base modifications, such as 2-thiouridine, N6-methyl-adenosine (N6-methyladenosine; m^6^A), and N1-methyl-pseudo-uridine (N1-methylpseudouridine; m^1^ψ) [234], due to changes in the secondary structure of IVT-mRNA, limiting its recognition by TLRs and nucleases. However, the downside of these modifications is the potential reduction in translation capability.

### 5.3. Non-Viral IVT-mRNA System for Mediated Transfer and Expression

Delivery systems, in terms of IVT-mRNA technology, play a crucial role in the efficient in vivo delivery of the IVT-mRNA and the further expression of the desired therapeutic protein into the recipient cells’ cytoplasm. IVT-mRNAs are characterized by high molecular weight, anionic charge, and hydrophilicity, which make it difficult for them to enter cells without a delivery mechanism mediating their internalization [235].

As previously mentioned, IVT-mRNA must overcome several obstacles in order to achieve its therapeutic goal in the target-site: the RNases, which are abundant in all biological fluids, and the hampering of passive diffusion by the cell membrane (permitted only for molecules that are smaller than 1000 Da) [236]. According to Sahin et al., if 10,000 molecules of naked IVT-mRNA enter the human body, only one would be delivered into cells [168]. Moreover, Schlake et al. reported that the intracellular half-life of naked IVT-mRNA is ∼7 h [237]. These issues will be bypassed by an ideal carrier that would shield the therapeutic IVT-mRNA, prevent the immune detection, and, if possible, offer a targeted delivery to the site of interest [238]. The delivery system is needed to generate efficient complexation, prolonged circulation for systemic delivery, prevent IVT-mRNA degradation by harmful nucleases [239], facilitate the cellular uptake, and finally promote the endosomal escape and efficient protein expression into host cells [240].

#### 5.3.1. Physical Approaches for the Delivery of IVT-mRNA

Several physical methods that have been extensively used for the delivery of pDNA were also used for delivering naked IVT-mRNA. Starting with ex vivo manipulation, the direct injection of IVT-mRNA into recipient cells can be mediated by microneedles [241,242] or the “gene gun” method, where the IVT-mRNA is shot into the target cell. Regarding in vivo delivery of naked IVT-mRNA through i.v. administration, RNases and the innate immune system will prevent delivery into target cells almost immediately [243]. However, during vaccination, i.d., i.m., s.c., or i.n. injections [244] are preferred to trigger immune system activation [245]. To overcome these issues, more physical methods have been developed, such as microporation (variation of electroporation) [246,247,248], electroporation, iontophoresis [249], sonophoresis [250], as well as nucleofection [251,252] and magnetoporation. Electroporation has been extensively used in cancer immunotherapy for the ex vivo genetic processing of autologous DCs with the antigen-encoding IVT-mRNAs before the re-injection of DCs back to the patient [253].

#### 5.3.2. Nanocarriers for the Delivery of IVT-mRNA

Today, the most advanced clinical application of IVT-mRNA technology is the vaccination against SARS-CoV-2, namely the Pfizer/BioNTech Comirnaty vaccine (BNT162b2) (listed for emergency use on 31 December 2020 [WHO, 2020]) and the Moderna COVID-19 vaccine (mRNA-1273) (listed for emergency use on 30 April 2021 [WHO, 2021]). IVT-mRNA vaccines use lipid-nanoparticles as the nanocarrier for the delivery of the mRNA, encoding the spike protein of the virus. More specific, in BNT162b2 the ionizable lipid ALC-0315 ((4-hydroxybutyl) azanediyl)bis (hexane-6,1-diyl)bis(2-hexyldecanoate)) and in mRNA-1273 the SM-102 (heptadecan-9-yl 8-((2-hydroxyethyl) (6-oxo-6-(undecyloxy) hexyl) amino) octanoate) were used [254].

Nanocarriers used for IVT-mRNA delivery differ in composition, size, shape, and physicochemical properties. These nanocarriers are mostly made of organic, biocompatible, or synthetic materials. Each kind of carrier should prevent the nucleic acid from degrading, thereby helping the transfection process, but it should also be as non-toxic and immune-suppressive as possible. The orchestration of the IVT-mRNA release profile would be another desirable trait of carriers, providing an enhanced pharmacokinetic profile, less toxicity in healthy organs/tissues, and increased IVT-mRNA circulation time in the blood [235]. Lipidic, polymeric, polypeptidic systems, dendrimers, gold nanoparticles, and hybrid systems are some of the most well-known and often utilized nanocarrier systems [235]. The first class of synthetic lipid carriers for IVT-mRNA was composed of cationic liposomes with permanent charges. Liposomes are spherical vesicles made of one or more phospholipid layers. Typically, materials with polar head groups and non-polar tails are used to generate the vesicle, which contains an aqueous core carrying the target gene. The interactions between these hydrophobic and hydrophilic groups promote the creation of vesicles. Electrostatic interactions allow positively charged cationic lipids to assemble with negatively charged IVT-mRNA to create the multilayer cystic complex known as a lipoplex (LP). RNases cannot readily reach the IVT-mRNA enclosed in the LP, allowing for its delivery without destruction. The fact that cationic lipids are also positively charged under physiological circumstances means that they are more likely to interact with other negatively charged molecules in biological fluids.

Malone et al. were the first to report the use of DOTMA (N-[1-(2,3-dioleyloxy)propyl]-N,N,N-trimethyl-ammonium chloride) [255] in combination with DOPE (dioleoyl-phosphatidyl-ethanol-amine) for the transfection of the luciferase IVT-mRNA [256]. Derived from DOTMA, another synthetic lipid DOTAP (1,2-dioleoyl-3-trimethyl-ammonium-propane) was then used [257] for the IVT-mRNA delivery. However, persistent charges on lipids frequently result in cell membrane deterioration and serum protein adsorption. Their exploitation in vivo is severely constrained by their toxicity and immunogenicity [258]. Lipoplexes’ stability has been demonstrated to be enhanced by the addition of non-charged polymers, such as polyethylene glycol (PEG), polyethyleneimine (PeI), poly-L-lysine, dendrimers, DEAE-dextran, poly(β-amino ester) (PBAE), and chitosan [259,260].

Furthermore, ionizable amine-bearing lipids may alter their charge depending on the pH, in contrast to permanently cationic lipids, which is advantageous for IVT-mRNA delivery since it results in less toxicity and more effective endosomal escape through the “flip-flop” or “proton sponge” process. An ionizable lipid is the 1,2-dilinoleyloxy-3-dimethylaminopropane (DLinDMA) lipid, which was used for the delivery of self-amplifying mRNA [257,261,262] as well its optimized form, DLin-MC3-DMA [263,264,265,266].

Another category of IVT-mRNA nanocarriers is the cationic polymers, both nondegradable and degradable, synthesized by natural or synthetic materials. Their function is quite similar to lipid nanocarriers; however, polymers offer protection from RNase degradation, promote IVT-mRNA cellular uptake, and facilitate endosomal escape [244]. Among cationic polymers are PEI, polyamidoamine (PAMAM) dendrimer, polysaccharide, PAsp(DET) [267], PBAE [268], and poly(amine-co-ester) (PACE) terpolymer [269].

IVT-mRNA has been shown to be successfully delivered also by using anionic polymers, such as the polylactic-co-glycolic acid (PLGA) [270]. Cationic lipid materials were added to anionic polymers to develop lipid–polymer hybrid formulations since the latter could not effectively encapsulate the negatively charged mRNA molecules.

### 5.4. Peptides for IVT-mRNA Delivery and Expression

Peptides can also be considered as a very promising category for IVT-mRNA carriers. The most appealing candidates would logically be positively charged peptides, mainly containing lysine (K) and arginine (R) residues, to be complexed with the mRNA through electrostatic interactions. Such an example is protamine, a cationic peptide enriched in arginine, which was spontaneously complexed with IVT-mRNA and facilitated its cellular uptake [271]. In fact, CureVac used protamine to protect mRNA from RNase degradation, calling this technology RNActive, which acts also as an initiator of immunity and an adjuvant. Nowadays, protamine is the only peptide-based IVT-mRNA nanocarrier undergoing clinical evaluation [272,273,274].

PTDs also bear a great potential to be exploited as IVT-mRNA delivery vehicles. They could make ideal delivery vehicles because they have low charge densities and can transduce membranes and further allow endosomal escape [275]. In 1999, the first successful transduction of a fusion recombinant protein with a PTD to various tissues in mice was published [180]. Subsequently, it was shown that PTDs can also intracellularly transduce nucleic acids [276] and nanomaterials in various cell lines. From then on, PTDs have been exploited in gene therapy approaches, with enhanced delivery efficiency. Several peptide-based intracellular delivery systems have been reported in the literature, both alone and in combination with other materials such as polymers.

Most studies use positively charged PTDs that may readily cross negatively charged cell membranes through endocytosis or independent of endocytosis (direct penetration) processes and without causing cytotoxicity [277]. Peptide-based nanoparticles (PBNs) are created by fusing a PTD or a fused-PTD (for example, PEGylated) with nucleic acids (NA), such as pDNA, IVT-mRNA, siRNA, or antisense oligonucleotide (ASO) [278,279,280,281,282,283,284,285,286], at a certain molar or charge proportion. Peptides and NAs are combined to form nanoparticles that can self-assemble into PBNs, multi-grafted PBNs, or impending micelle-like PBNs. The NAs are always enclosed by the PBNs, which are thought to range in size from 60 nm to 150 nm. From there, concerted translocation or endocytosis-dependent mechanisms may be used for cellular internalization [283].

Firstly, the PTDs were shown to deliver NAs via the chemistry of peptide nucleic acids (PNAs) [287], which are synthetic polymers, such as DNA or RNA. Synthetic oligomeric PNAs have been used in recent years in molecular biology procedures, diagnostic assays, and antisense therapies. In PNAs, the deoxyribose phosphate backbone is replaced by an N-(2-aminoethyl)glycine linkage, and the nucleobases are linked via a methylene-carbonyl bond to the glycine amino group. Since PNAs have a peptide-like neutral backbone, it was easy to generate the PTD–PNA complexes by coupling with the peptide (PTD) in various ways, such as continuous solid-phase synthesis, maleimide coupling, and/or through an ester or disulfide bond. Several PTDs enriched in lysine and arginine have been shown to enhance PNA uptake and activity, especially when the PNAs were antisense RNAs. Bendifallah et al. performed screening studies of various well-known PTDs (Transportan, oligo-arginine (R7-9), pTAT, Penetratin, KFF, SynB3, etc.) for their efficacy in mediating PNA intracellular uptake via covalent binding [288,289].

However, regarding the delivery of IVT-mRNA [214], there are limited studies using PTDs as delivery vehicles through covalent and non-covalent conjugation approaches. The first approach uses chemical linkers to create non-covalent bonds between the PTD and its cargo, whereas the second method relies on electrostatic self-assembly of non-covalent bonds. Sulfosuccinimidyl suberate linkage, carbodiimide coupling, and thiol-amine coupling have all been used to create covalent interactions. Electrostatic, metal-affinity, and biotin–streptavidin interactions are examples of non-covalent interactions [290].

High delivery efficiency and a theoretically stable connection of the PTD to NAs are guaranteed by the covalent conjugation. Covalent conjugation may be a time-consuming and a difficult procedure. Additionally, there is the risk of chemically changing the cargo to achieve covalent attachment to the PTD, which may jeopardize their functioning. The benefits of a non-covalent interaction between the PTD and the cargo, on the other hand, are simplicity of use, ease of manufacture, flexibility regarding the cargo’s composition, and preservation of cargo functioning. In order to deliver siRNA, plasmids, and splicing-correction oligonucleotides with enhanced transfection and biological effectiveness, non-covalent techniques have been adopted more often [290,291].

In an interesting study, three PTDs were used for IVT-mRNA delivery, namely RALA, LAH4, and LAH4-L1, combined through non-covalent interactions, where all three peptides successfully mediated the intracellular delivery of IVT-mRNA in DCs through phagocytosis and clathrin-dependent endocytosis. The optimal delivery efficiency of the IVT-mRNA was demonstrated by the PTD LAH4-L1 [292].

One of the first commercially available peptides used was PepFect14, which was complexed with eGFP (eGreen fluorescent protein) IVT-mRNA through electrostatic interactions [293]. Furthermore, the same group published a study with an effective IVT-mRNA delivery system using a “truncated” (D-amino acid based) protamine, the Xentry-protamine [294]. Another PTD, the peptide p5RHH (modified membrane lytic protein melittin), was successfully employed for the intracellular transduction of the near-infrared fluorescent protein (niRFP)-encoding mRNA through a non-covalent, self-assembly manner [295]. Through non-covalent conjugation, Mela-A was also used for complexation with dsRNA to activate DCs [296]. Finally, R9 peptide (CRPPR-R9) mediated the intracellular transduction of the IVT-mRNA of transcription factors to reprogram cardiac fibroblasts into cardiomyocytes [297].

All the above approaches are categorized as non-covalent conjugation strategies of peptides to IVT-mRNA. However, there have been reported some covalent approaches that are advantageous due to the stable nature of the complexation owing to the physicochemical characteristics of the bonds and the peptide itself. The peptides used for this approach are usually amphipathic in nature owing hydrophilic and hydrophobic domains [298,299]. The main idea started from the mRNA display method for the in vitro screening of peptide libraries [300,301,302,303], where a peptide is covalently conjugated with the 3’-terminus of its own mRNA, forming a peptide–mRNA hybrid. This idea was also reported with the random co-polymer pHDPA, which was conjugated to the IVT-mRNA of ovalbumin through complexation post PEG-peptide modification, followed by click-chemistry conjugation with the anionic peptide GALA and cross-linking for stabilization, leading to the successful cellular uptake of this complex into DCs [304].

#### PTD-Mediated PRT: Transduction of the IVT-mRNA of Sco2 Mitochondrial Protein

Our team exploited an established PTD previously used for the intracellular transduction of DNA nanoparticles, proteins, siRNAs, and other molecules, that is, the 6aa peptide PFVYLI, for the intracellular delivery of the IVT-mRNAs. PFVYLI is a shorter form of the peptide C105Y, a synthetic PTD derived from the residues 359–374 of 1-antitrypsin [305,306,307,308]. PFVYLI is a hydrophobic peptide with neutral surface charge, leading to the need for a linker to be covalently conjugated with the negatively charged IVT-mRNA. Using this hydrophobic PTD peptide covalently coupled to any IVT-mRNA through a unique covalent chemical reaction, our team, i.e., Miliotou et al., patented a methodology to generate a universal delivery platform. Puromycin, which is conjugated to the PFVYLI through an amide bond, acts as a linker to the IVT-mRNA in this unique chemical reaction [309]. The procedure describing the unique chemical process for producing PTD (stands for the PTD peptide PFVYLI)-IVT-mRNAs is now the subject of an international patent application with publication number WO2021/094792A1, entitled “Method for the development of a delivery platform to produce deliverable PTD-IVT-mRNA therapeutics” [309]. The number of the Greek patent is 1010063, granted to our group until 2039 by the National Industrial Property Organization (OBI), while an application to the European Patent Office with the number ΕΡ20823912.9 was recently submitted.

Firstly, the in vitro transcription template was cloned in a pcDNA3.1(+) plasmid, which included the CDS of the full-length *SCO2* gene plus the upstream 5′-UTR of mouse β-globin (with a strong Kozak sequence) and downstream the 3′-UTR of human β-globin for purposes of achieving greater stability of IVT-mRNA. In the next step, the in vitro transcription of the *SCO2* mRNA was performed. It is a quite sensitive process that requires delicate handling, sterile conditions, and cleanliness. Successful transcription of the IVT-mRNA of *SCO2* was verified by gel electrophoresis for size-based separation of mRNAs as well by two-step RT-PCR use corresponding to gene-specific primers, designed inside the UTRs to discriminate the IVT-mRNA from the endogenous recipient mRNA. Subsequently, the coupling reaction of the selected PTD with the IVT-mRNA was carried out. To check for successful ligation, “band shift assay” were performed, where the PTD caused a delay in the migration of IVT-mRNA through the polyacrylamide gel. In parallel, a control (non PTD) peptide was used as a negative marker in all the transduction experiments. The control peptide-IVT-mRNA of *SCO2* also showed slower transposition in the band shift assay compared to the naked IVT mRNA of *SCO2.* To investigate the nature of the bonds between the PTD and the IVT-mRNA, NMR analysis of the PTD-IVT-mRNA of *SCO2* was also performed and was compared to the proton spectrum of the free PTD. A shift of the amide protons was also observed as double peaks. Furthermore, the intensity of the peaks decreased in the presence of (deuterated) heavy water (D_2_O), due to the exchange of hydrogens to deuteriums, indicating that the changes in the chemical environment due to the conjugation reaction led to a differentiation of the magnetic environment. Thus, by both the band shift evaluation assay and the NMR analysis, it was shown that the PTD (PFVYLI) was effectively conjugated to IVT-mRNAs, producing the PTD-IVT-mRNA of *SCO2*.

The conjugated PTD was found to offer protection to the IVT-mRNA from the nuclease digestion during the structural stability assays of the produced PTD-IVT-mRNA of *SCO2*. The PTD-IVT-mRNA remained structurally intact under the cell culture conditions employed, even in 10% fetal bovine serum and in the presence of RNase A at 37 °C and for different time intervals. PTD-IVT-mRNA of *SCO2* effectively transduced K562 cells, such as a cell-line model, without any cytotoxicity, showing increased *SCO2* translation from the first 30 min of incubation, increasing over time, up to 96 h post-transduction. The IVT-mRNA of *SCO2* transduced was also detected, showing increased mRNA levels intracellularly up to 96 h. PTD-IVT-mRNA of *SCO2* was also effectively detected and showed increased translation into the Sco2 protein and into primary *SCO2*/COX-deficient fibroblasts from 30 min to 96 h post transduction. More specifically, 72 h after transduction, Sco2 protein was expressed with up to ∼7-fold increase compared to Sco2 levels in untreated *SCO2*/COX-deficient fibroblasts. Furthermore, uptake of the PTD-IVT-mRNA of *SCO2* was shown to be mediated via a clathrin-mediated endocytic pathway.

Finally, histochemical analysis was performed in primary fibroblasts derived from the *SCO2*/COX-deficient patient (30% residual activity compared to control/healthy fibroblasts) [66], which were transduced with the PTD-IVT-mRNA of *SCO2*, to evaluate the phenotypic complementation by restoring COX activity. Transduced *SCO2*/COX-deficient fibroblasts showed increased staining from 30 min post transduction up to 96 h, indicating the sufficient functionality and deficient phenotype’s restoration. 

Thus, the PTD-IVT-mRNA platform was successfully developed as a therapeutic approach, which ensures IVT-mRNA protection in harmful, RNase-rich environments and achieves sufficient transduction rates in our cell model of primary fibroblasts derived from the *SCO2*/COX-deficient patient.

Overall, these results highlight the potential of using this innovative PTD-IVT-mRNA delivery approach in preclinical/clinical trials as a safe PRT for mitochondrial disorders, offering a promising platform for efficient gene expression [309].

## 6. Conclusions

The current review focuses on mitochondrial genetic disorders in order to present recent innovative approaches relying on PRT as well as IVT-mRNA-based therapeutic methodologies. In particular, this review presents the structural as well as the functional characteristics of human mitochondria, i.e., the organelles playing the central role in cell respiration, bioenergetics, and metabolism. Following that, a thorough discussion is included based on the mitochondrial genetic disorders that are attributed mainly to genetic mutations/pathogenic variants affecting the structure and functions of the mitochondrial protein Sco2 that lead to both severe Sco2 deficiency and COX deficiency. This severe disorder affects all cell types in the human body and is translated as fatal infantile cardioencephalomyopathy. Unfortunately, there is no therapy available at present for this severe monogenic mitochondrial disorder since mitochondrial disorders have proven intractable to treatment so far. 

The data included in this article present, in considerable detail, the latest development of methods and innovation to address the question of whether PRT can offer safe and effective therapy for this mitochondrial disorder. Within the frame of this innovative therapy, the review presents and focuses on PTD technology as a new promising therapeutic modality. At the beginning of the research work from our own laboratory, this approach was designed to facilitate the delivery of the full-length recombinant Sco2 protein, fused with the TAT as PTD, inside the cells. Intracellular fusion Sco2 protein is translocated to the mitochondria to be processed by the MPP, giving rise to the mature protein, and the following insertion in the mitochondrial inner membrane is necessary for the assembly of COX and restoration of the COX activity. Moreover, our work has shown that this recombinant fusion Sco2 protein administered in healthy mice was identified in the mitochondria of all tissues examined, including brain, muscle, and heart, although it was to a lesser extent than in other tissues such as liver and spleen. Alternatively, we developed a patented, novel, PTD-mediated, in vitro transcribed (IVT)-mRNA delivery platform to mediate the transduction and express the IVT-mRNA of full-length *SCO2* that is translated—upon its entrance inside the cells—to the corresponding Sco2 protein, followed by its translocation to mitochondria and recovery of COX activity, thus reversing the phenotype of this monogenic mitochondrial disorder. Both approaches using the recombinant fusion Sco2 protein or the PTD-IVT-mRNA of *SCO2* provide a potential PRT for the treatment of fatal infantile cardioencephalomyopathy due to *SCO2* deficiency. Therefore, our studies on PRT with PTD-mediated delivery of either Sco2 protein or, alternatively, IVT-mRNA of *SCO2* are considered as novel in this field.

In conclusion, these approaches can serve as a model for the treatment of other monogenic mitochondrial disorders, that are frequently resulted from dysfunctional or depleted mitochondrial proteins and can confer as a prime example of a disease area in which PRT can be applied.

## Figures and Tables

**Figure 1 pharmaceutics-15-00286-f001:**
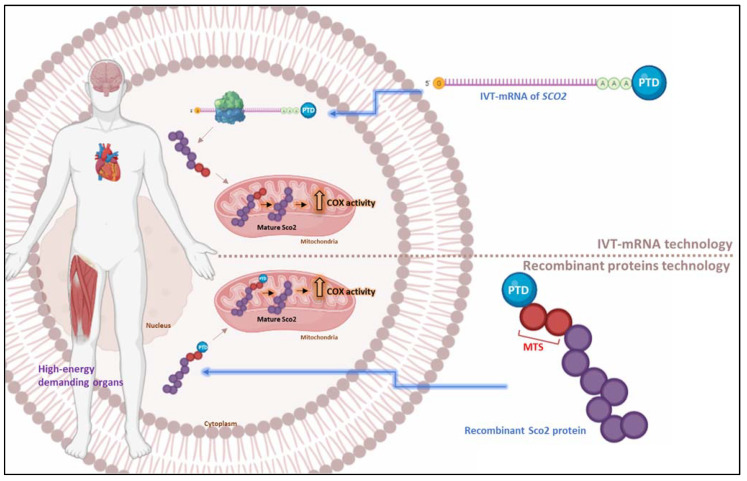
The exploitation of PTD technology for the efficient expression of the Sco2 protein into cells derived from a *SCO2*/COX-deficient patient. PTDs can mediate the efficient intracellular transduction of recombinant full-length Sco2 protein or the transduction of the IVT-mRNA of *SCO2*, which will be translated to Sco2 protein into cells’ cytoplasm. Notably, by both approaches, recombinant Sco2 will be delivered inside the mitochondria through its mitochondrial targeting signal peptide and be processed by mitochondrial processing peptidase to give the mature Sco2, thus facilitating the recovery of COX activity.

**Table 1 pharmaceutics-15-00286-t001:** Reported Sco2 mutations and associated pathologies.

*SCO2* Genotype	Clinical Outcome	Affected Individual(s)	References *
Q53X/E140K	Fatal infantile cardioencephalomyopathy with COX deficiency, neurological symptoms with lactic/metabolic acidosis, respiratory difficulties	6	Papadopoulou L.C. et al., 1999 [66]; Tay S.K.H. et al., 2004 [110]; Vesela K. et al., 2004 [113]; Pronicki M. et al., 2010 [103]; Pronicka E. et al., 2013 [102]
E140K/S225F	Fatal infantile cardioencephalomyopathy with COX deficiency	1	Papadopoulou L.C. et al., 1999 [66]
E140K/R171W	Fatal hypertrophic cardiomyopathy (HCMP), seizures, muscle hypotonia (MH), respiratory insufficiency	1	Jaksch M. et al., 2000 [97]
R90X/E140K	Fatal hypertrophic cardiomyopathy (HCMP), seizures, muscle hypotonia (MH), respiratory insufficiency	2	Jaksch M. et al., 2000 [97]
E140K/E140K	Delayed infantile onset of cardiomyopathy and neuropathy, laryngeal inspiratory stridor, infantile SMA-like/Leigh-like picture	39	Jaksch M. et al., 2001 [122]; Vesela K. et al., 2004 [113]; Bohm M. et al., 2006 [118]; Pronicki M. et al., 2010 [103]; Pronicka E. et al., 2013 [102]
10 bp duplication (1302–1311)/E140K	Prominent spinal cord involvement mimicking spinal muscular atrophy (Werdnig–Hoffmann disease)	1	Salviati L. et al., 2002 [106]
E140K/L151P	Hypertrophic cardiomyopathy and encephalomyopathy	1	Sacconi S. et al., 2003 [105]
C133S/E140K	Neonatal hypotonia with a spinal muscular atrophy (SMA) type 1 phenotype	1	Tarnopolsky M.A. et al., 2004 [109]
1518delA/E140K	Cytochrome *c* oxidase deficiency and a Werdnig–Hoffmann disease phenotype	2	Bohm M. et al., 2006 [118]; Vesela K. et al., 2008 [114];
Hemizygosity 16 bp deletion within the intron_E140K	Early onset rapidly progressive, fatal cardiomyopathy	1	Leary S.C. et al., 2006 [100]
E140K/V160G	Cytochrome *c* oxidase deficiency, fatal infantile cardioencephalomyopathy	1	Knuf M. et al., 2007 [99]
W36X/E140K	Fatal infantile cardioencephalomyopathy	2	Verdijk R.M. et al., 2008 [112]
G193S/G193S	Fatal infantile cardioencephalomyopathy	1	Mobley B.C. et al., 2009 [101]
E140K/M177T	Classical SMA or SMA-like picture, laryngeal inspiratory stridor, milder encephalopathic	4	Pronicki M. et al., 2010 [103]; Pronicka E. et al., 2013 [102]
E140K/w/t	Respiratory failure, artificial ventilation, hypotony, high-grade myopia	4	Pronicki M. et al. 2010 [103]; Tran-Viet K.N. et al. 2013 [111]
19 bp insertion at position 17 (17INS19bp)/ E140K	Failure to thrive, muscular hypotonia, hypertrophic cardiomyopathy, and lactic acidemia, totally absent of COX activity	1	Joost K. et al., 2010 [98]
12 bp deletion (c.1519_1530del)/ E140K	Cardioencephalomyopathy, stridor, neuropathy	1	Gurgel-Giannetti J. et al., 2013 [96]
Q53X/w/t	High-grade myopia	7	Tran-Viet K.N. et al., 2013 [111]
R114H/w/t	High-grade myopia	1	Tran-Viet K.N. et al. 2013 [111]
p82delK/E140K	Progressive encephalopathy, cardiomegaly, spinal muscular atrophy, COX deficiency	1	Pronicka E. et al., 2013 [102]
W75R/E140K	Neonatal cardiomyopathy, muscle weakness	1	Pronicka E. et al., 2013 [102]
E140K/T241X	Neonatal cardiomyopathy, muscle weakness, Leigh disease	1	Pronicka E. et al., 2013 [102]
V160A/P233T	Fatal hyperthermia and metabolic acidosis	1	Sambuughin N. et al., 2013 [107]
A259V/w/t	High-grade myopia	1	Tran-Viet K.N. et al., 2013 [111]
D223N/87 kb deletion on chr.22	Severe hypotonic syndrome, failure to thrive, divergent strabismus and ataxia, regression of psychomotor development	1	Vondrackova A. et al., 2014 [115]
R112W/w/t	Early-onset high myopia	1	Jiang D. et al., 2014 [119]
R120W/w/t	Early-onset high myopia	1	Jiang D. et al., 2014 [119]
A97V/w/t	Extreme myopia	1	Wakazono T. et al., 2016 [117]
D135G/R171Q	Early-onset axonal Charcot–Marie–Tooth disease associated with cellular copper deficiency	1	Rebelo A.P. et al., 2018 [104]
E140K/P169T	Early-onset axonal Charcot–Marie–Tooth disease associated with cellular copper deficiency	1	Rebelo A.P. et al., 2018 [104]
D173V/w/t	Non-syndromic high myopia	1	Cai X.B. et al., 2019 [120]
A201P/w/t	Non-syndromic high myopia	1	Cai X.B. et al., 2019 [120]
I221V/w/t	Non-syndromic high myopia	1	Cai X.B. et al., 2019 [120]
R255W/R255W	Cerebellar ataxia, progressive peripheral axonal neuropathy and long survival	2	Barcia G. et al., 2019 [94]
p82delK/w/t	Non-syndromic high myopia	1	Zheng Y.H et al., 2021 [121]
R60Q/G193S	Adult cerebellar ataxia, axonal neuropathy, and sensory impairments	1	Rucheton B. et al., 2021 [116]
G121R/G121R	Early-onset axonal Charcot–Marie–Tooth disease	2	Gangfuß A. et al., 2022 [95]

***** Sco2 mutants are shown in chronological order based on their first report.

**Table 2 pharmaceutics-15-00286-t002:** Clinical trials using gene therapy or recombinant protein therapy and PRT for human severe genetic mitochondrial disorders, according to www.clinicaltrials.gov.

Condition/ Disease	Title	Interventions	Phase	Clinical Trial
Leber Hereditary Optic Neuropathy (LHON) disease	Gene Therapy Clinical Trial for the Treatment Of Leber’s Hereditary Optic Neuropathy (GOLD)	Drug: NR082 injection Device: sham injection	Phase 2 Phase 3	NCT04912843
	Safety Study of an Adeno-associated Virus Vector for Gene Therapy of Leber’s Hereditary Optic Neuropathy	Drug: injection of scAAV2-P1ND4v2 (low–higher)	Phase 1	NCT02161380
	Efficacy & Safety Study of Bilateral IVT Injection of GS010 in LHON Subjects Due to the ND4 Mutation for up to 1 Year	Genetic: GS010 Drug: placebo	Phase 3	NCT03293524
	A Single Intravitreal Injection of rAAV2-ND4 for the Treatment of Leber’s Hereditary Optic Neuropathy	Drug: rAAV2-ND4	Phase 2 Phase 3	NCT03153293
	RESCUE and REVERSE Long-term Follow-up	Genetic: GS010 Other: sham	Completed (2022)	NCT03406104
	Safety Evaluation of Gene Therapy in Leber Hereditary Optic Neuropathy (LHON) Patients	Genetic: GS010	Completed (2020)	NCT02064569
	REALITY LHON Registry	Other: patient-reported outcomes (PROs)	Completed (2020)	NCT03295071
	Efficacy Study of GS010 for the Treatment of Vision Loss up to 6 Months From Onset in LHON Due to the ND4 Mutation	Biological: GS010 Device: sham intravitreal injection	Completed (2020)	NCT02652767
	Safety and Efficacy Study of rAAV2-ND4 Treatment of Leber Hereditary Optic Neuropathy (LHON)	Drug: rAAV2-ND4	Completed (2016)	NCT01267422
Friedreich Ataxia	Gene Therapy for Cardiomyopathy Associated With Friedreich’s Ataxia	Genetic: low–high dose LX2006	Phase 1 Phase 2	NCT05445323
	Phase IA Study of AAVrh.10hFXN Gene Therapy for the Cardiomyopathy of Friedreich’s Ataxia	Biological: AAVrh.10hFXN, serotype rh.10 adeno-associated virus (AAV) gene transfer vector expressing the cDNA coding for human FXN Drug: Prednisone	Phase 1	NCT05302271
	Multiple Ascending Dose Study of CTI-1601 Versus Placebo in Subjects With Friedreich’s Ataxia	Biological: CTI-1601 (a recombinant fusion protein intended to deliver human frataxin into the mitochondria of patients with Friedreich’s ataxia)	Phase 1	NCT04519567

## Data Availability

Not applicable.

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
