# Peer review of "Protein Transduction Domain-Mediated Delivery of Recombinant Proteins and In Vitro Transcribed mRNAs for Protein Replacement Therapy of Human Severe Genetic Mitochondrial Disorders: The Case of Sco2 Deficiency"

_pharmaceutics, 2023, doi:10.3390/pharmaceutics15010286_

Round 1
Reviewer 1 Report
In the submitted manuscript (ID 2102283) the authors describe the recent innovative approaches for Protein Replace Therapy (PRT), whose goal is to replace dysfunctional or not expressed proteins. PRT could be mediated by Protein Transduction Domain (PTD) and by new mRNA-based therapy. The authors focused on mitochondrial genetic disorders, severe diseases with no available therapy. In particular, the authors illustrate their results for the recovery the SCO2 deficiency, by administration of full-length recombinant SCO2 as protein or as mRNA.
The manuscript is well organized and well written. However, some minor revisions are necessary before final acceptance of the manuscript.
Minor revisions:
In my opinion the manuscript could be improved and could be more interesting for readers if one or more tables were added summarizing the clinical trials or approved therapies using gene therapy and PRT based on PTD or IVT-mRNA technologies described in the text.
Line 221: The references Protasoni M., Zeviani M. DOI: 10.3390/ijms22020586 and Marra et al DOI: 10.3390/biom11111633 should be added.
Line 317: The reference Brischigliaro M., Zeviani M. DOI: 10.1016/j.bbabio.2020.148335 should be added.
Author Response
Point 1: In the submitted manuscript (ID 2102283) the authors describe the recent innovative approaches for Protein Replace Therapy (PRT), whose goal is to replace dysfunctional or not expressed proteins. PRT could be mediated by Protein Transduction Domain (PTD) and by new mRNA-based therapy. The authors focused on mitochondrial genetic disorders, severe diseases with no available therapy. In particular, the authors illustrate their results for the recovery the SCO2 deficiency, by administration of full-length recombinant SCO2 as protein or as mRNA.
The manuscript is well organized and well written.
Response 1: We are delighted for this comment by Reviewer #1 and it is very much appreciable. We would also like to thank Reviewer #1 for his/her constructive comments that significantly improved our manuscript.
Point 2: In my opinion the manuscript could be improved and could be more interesting for readers if one or more tables were added summarizing the clinical trials or approved therapies using gene therapy and PRT based on PTD or IVT-mRNA technologies described in the text.
Response 2: As the Reviewer indicated we proceeded in adding Table 2, which includes clinical trials using gene therapy or recombinant protein therapy and PRT for human severe genetic mitochondrial disorders (Leber Hereditary Optic Neuropathy (LHON) disease and Friedreich Ataxia). Please find the text addition in lines 477-478 and Table 2 in lines 494-496.
Point 3: The references Protasoni M., Zeviani M. DOI: 10.3390/ijms22020586 and Marra et al DOI: 10.3390/biom11111633 should be added.
Response 3: We are grateful for this comment, and we proceeded in the addition of these interesting references.
Point 4: Line 317: The reference Brischigliaro M., Zeviani M. DOI: 10.1016/j.bbabio.2020.148335 should be added.
Response 4: As the Reviewer indicated we proceeded in the addition of this reference, but in line 310.
Furthermore, we made some minor additions/corrections throughout the text:
Line 7: asterisk (*) nearby the name Lefkothea C. Papadopoulou1
Line 14: * Added “Author to whom correspondence should be addressed”
Line 16: Deleted “and”
Line 89: Deleted “Penetratin:Pen”
Line 253 and 256: Entered empty lines
Table 1:
- Converted “patients no.” to “affected individual(s)”
- Converted all “;” to “,”
- In clinical outcome column:
- Q53X / E140K - changed to “neurological symptoms with lactic/metabolic acidosis, respiratory difficulties”
- E140K / R171W - changed to “hypertrophic”
- R90X / E140K - changed to “hypertrophic”
- E140K / E140K - added “picture”
- C133S / E140K- deleted full stop
- 1518delA /E140K - “c” in italics
- E140K / V160G - “c” in italics, corrected to “deficiency”
- 19bp insertion at position 17 (17INS19bp) / E140K - deleted full stop
- 12 bp deletion (c.1519_1530del) / E140K - added “stridor, neuropathy”
- V160A / P233T - deleted “MH (Malignant Hyperthermia)”
- D223N / 87kb deletion on chr.22 - added “ataxia”, deleted “his”
Line 341: Corrected to “Sco2 mutants are shown in chronological order based on their first report.”
Line 365: “e.g.,” changed to “i.e.”
Line 386: deleted “e.t.c.”
Line 406: added a comma (“lentiviruses,”)
Line 508: deleted “e.t.c.”
Line 562: “i.p.” no italics
Line 582: added a parenthesis
Line 592: “e.g.,” changed to “i.e.”
Lines 639 and 641: “i.p.” no italics
Line 693: changed “Again, by using” to “By using, again,”
Line 772: changed to “Hoerr I.”
Line 784: changed to “Karikó K.”
Line 940: added a comma (“biocompatible,”)
Line 1137: changed to “preclinical”

Reviewer 2 Report
It was a nice review paper about the application of PRD for the delivery of mRNAs and recombinant proteins for the treatment of human severe genetic mitochondrial disorders. Here are some comments on this study that should be considered before publication:
1- It seems there are some hyperlinks in the abstract. Please remove them.
2- Please use the same font for the text in the whole text.
3- “via” in line 195 shouldn’t be in italic. The same for “(oral, i.v., i.a., i.m.)” in line 388.
4- “A brief description of the technology of IVT-mRNA therapeutics follows in the next chapter.” it is better to write this sentence as this “A brief description of the technology of IVT-mRNA therapeutics is described in the following section.”.
Author Response
Point 1: It was a nice review paper about the application of PRD for the delivery of mRNAs and recombinant proteins for the treatment of human severe genetic mitochondrial disorders.
Response 1: We would like to thank Reviewer #2 for finding our review paper nice, his/her comments are really appreciated and gave us the opportunity to optimize our manuscript.
Point 2: It seems there are some hyperlinks in the abstract. Please remove them.
Response 2: We would like to apologize and we can ensure the Reviewer #2 that we removed the hyperlinks from the abstract.
Point 3: Please use the same font for the text in the whole text.
Response 3: We would like to thank the Reviewer #2 for this constructive comment, and we proceeded in keeping the same font (10) for the whole text.
Point 4: “via” in line 195 shouldn’t be in italic. The same for “(oral, i.v., i.a., i.m.)” in line 388.
Response 4: We would like to thank the Reviewer #2 for noticing this point and we proceeded in reversing from italics to normal, the words “via”, “i.p.” and “(oral, i.v., i.a., i.m.)”, in the whole text.
Point 5: “A brief description of the technology of IVT-mRNA therapeutics follows in the next chapter.” it is better to write this sentence as this “A brief description of the technology of IVT-mRNA therapeutics is described in the following section.”.
Response 5: We proceeded in correcting this point.
